



# Glacial history of Inglefield Land, north Greenland from combined in-situ [10]Be and [14]C exposure dating

Anne Sofie Søndergaard[1], Nicolaj Krog Larsen[1,2], Olivia Steinemann[3], Jesper Olsen[4], Svend Funder[2], David Lundbek Egholm[1], Kurt Henrik Kjær[2]

[1]Department of Geoscience, Aarhus University, Høegh Guldbergs Gade 2, 8000 Aarhus C, Denmark
[2] Globe Institute, University of Copenhagen, Øster Voldgade 5-7, 1350 Copenhagen K, Denmark
[3]Department of Physics, Institute for Particle Physics and Astrophysics, ETH Zürich, Otto-Stern-Weg 5, 8093 Zürich, Switzerland
[4]Department of Physics and Astronomy, Aarhus University, Ny Munkegade 120, 8000 Aarhus C, Denmark

*Correspondence to:* Anne Sofie Søndergaard (annesofie@geo.au.dk)

**Abstract.** Exposing the sensitivity of the Greenland Ice Sheet (GrIS) to Holocene climate changes is a key prerequisite for understanding the future response of the ice sheet to global warming. In this study, we present new information on the Holocene
glacial history of the GrIS in Inglefield Land, north Greenland. We use [10]Be and in-situ [14]C exposure dating to constrain the timing of deglaciation in the area and radiocarbon dating of reworked molluscs and wood fragments to constrain when the ice sheet retreated behind its present-day extent. The [10]Be ages are scattered ranging from c. 92.7 to 6.8 ka whereas the in-situ [14]C ages range from c. 14.2 to 6.7 ka. Almost half of the apparent [10]Be ages predate the Last Glacial Maximum and up to 89 % are to some degree affected by nuclide inheritance. Based on the few reliable [10]Be ages, the in-situ [14]C ages and existing
radiocarbon ages from Inglefield Land, we find that the deglaciation along the coast commenced c. 8.6-8.3 cal. ka BP in the western part and c. 7.9 ka in the central part, following the opening of Nares Strait and arrival of warm waters. The ice margin reached its present-day position c. 8.2 ka at the Humboldt Glacier and c. 6.7 ka in the central part of Inglefield Land. Radiocarbon ages of reworked molluscs and wood fragments show that the ice margin was behind its present-day extent from c. 5.8 to 0.5 cal. ka BP. After 0.5 cal. ka BP, the ice advanced towards its Little Ice Age position. Our results emphasize that
the slowly eroding and possibly cold-based ice in north Greenland makes it difficult to constrain the deglaciation history based on [10]Be ages alone unless it is paired with in-situ [14]C ages. Further, combining our findings with those of recently published studies reveals distinct differences between deglaciation patterns of northwest and north Greenland. Deglaciation of the land areas in northwest Greenland occurred earlier than in north Greenland and periods of restricted ice extent were longer, spanning middle and late Holocene. Overall, this highlights past ice sheet sensitivity towards Holocene climate changes in an area where
little information was available just a few years ago.



## 1 Introduction

Information about the glacial history of the Greenland Ice Sheet (GrIS) is important to constrain its sensitivity to past and
ongoing climate changes (Lecavalier et al., 2017; Larsen et al., 2018). Since the 1990s, mass loss from the GrIS has accelerated,
coinciding with atmospheric warming, and the ice sheet appears extremely sensitive to this warming, especially in north
Greenland, where the ablation area has expanded with 46 % (Khan et al., 2015; Noël et al., 2019). As a result, the relative
contribution to sea level rise from the north GrIS has increased significantly primarily through enhanced runoff as well as ice
discharge via calving and melting at the Humboldt Glacier front (Mouginot et al., 2019; Noël et al., 2019).

With the introduction of cosmogenic nuclide exposure dating, previously glaciated areas of Greenland have been
systematically targeted and >1000 [10]Be exposure ages have been published within the last two decades (Sinclair et al., 2016).
In consequence, the late glacial and Holocene glaciation history is well constrained in most areas of Greenland (Bennike and
Björck, 2002; Funder et al., 2011; Sinclair et al., 2016). However, there are still areas where the deglaciation chronology is
constrained by minimum limiting radiocarbon ages of mainly marine molluscs along the coast and where inland exposure ages

are unavailable (Bennike and Björck, 2002; Funder et al., 2011). This is particularly true for north Greenland including
Inglefield Land where the current knowledge is primarily based on studies from late 1960s and 1970s (Nichols, 1969; Tedrow,
1970).

Despite that, [10]Be exposure dating has shown to be an efficient tool for constraining the deglaciation of the GrIS, but the use
of this method is not without pitfalls. The method assumes that the measured [10]Be concentration was produced in one single

post-glacial exposure period without surface erosion, but a number of studies have demonstrated that this assumption does not
always hold. A particular challenge arises when subglacial bedrock erosion is too slow to remove [10]Be inventories produced
during earlier exposure periods, such as the previous interglacial. In such case, the resulting age is typically referred to as an
*apparent [10]Be exposure age* in acknowledgment of the fact that this age typically exceeds the true exposure age (Kelly et al.,
2008; Corbett et al., 2015; Farnsworth et al., 2018; Larsen et al., 2018; Søndergaard et al., 2019; Ceperley et al., 2020; Skov

et al., 2020). This problem of [10]Be nuclide inheritance emphasizes the need for new methods in order to thoroughly constrain
the glacial history in parts of Greenland where the cold subglacial thermal regime in many places dictates inefficient erosion
and widespread nuclide inheritance.

Here we use a combination of [10]Be and in-situ [14]C exposure dating of boulders and pebbles to overcome the problem of nuclide
inheritance and constrain the Holocene deglaciation history of the GrIS in Inglefield Land, north Greenland. In addition, we

use radiocarbon dating of reworked marine molluscs and wood fragments to constrain the Holocene timing of restricted ice
extent in the study area. Finally, we review and assess the glacial history in northwest and north Greenland with both local and
regional climate records in order to expand our knowledge of the long-term sensitivity of the GrIS to climate changes.



## 2 Study site and previous work

Inglefield Land is situated in north Greenland, between 78.2-79.1° N and 65.8-72.8° W and is bound to the south and east by the GrIS, to the west by Smith Sound and to the north by Kane Basin and Humboldt Glacier (Fig. 1). Humboldt Glacier drains c. 5 % of the GrIS into Nares Strait and has a varied velocity profile due to diverse bed topography and drainage networks (Rignot and Kanagaratnam, 2006; Hill et al., 2017; Livingstone et al., 2017). In addition to the marine terminating Humboldt Glacier, the land-based Hiawatha Glacier is present in the eastern part of Inglefield Land. Together with the GrIS, this glacier

overlie the newly discovered Hiawatha impact crater which makes the ice form a half circular structure, characterized by ice that flows faster than the rest of the ice margin terminating on land in Inglefield Land (Kjær et al., 2018).

The region is a high-Arctic desert, with low precipitation rates of c. 100-150 mm per year, falling mostly as snow (Blake et al., 1992; Dawes, 2004). The bedrock in the area is composed of Paleoproterozoic granite and gneiss, Late Proterozoic sedimentary and volcanic rocks and Lower Paleozoic sedimentary rocks and shelf carbonates with Quaternary deposits close

to the present-day ice margin (Dawes, 2004; Kolb et al., 2016). The relief is gently declining from 600-700 m a.s.l. close to the present-day ice margin towards the coast where meltwater channels and rivers cut the up to c. 400 m high plateaus composed of sedimentary rocks (Dawes, 2004).

During the Last Glacial Maximum (LGM) the GrIS and Inuitian Ice Sheet coalesced in Nares Strait and the ice flowed north- and southward from a saddle in Kane Basin (England et al., 2006). The southward flowing ice formed the Smith Sound Ice

Stream (England et al., 2006; Jennings et al., 2019) and it is believed to have extended to the 600 m depth contour in northern Baffin Bay (Funder et al., 2011). Radiocarbon ages reveal that deglaciation of Nares Strait initiated at the northern entrance c. 11 cal. ka BP and southern entrance c. 10 cal. ka BP (Bennike and Björck, 2002; Jennings et al., 2011). The final deglaciation and opening of the Nares Strait have been debated but recent off shore studies together with a few terrestrial studies place the collapse of the ice saddle in Kane Basin and opening of Nares Strait between c. 9-8 cal. ka BP (Georgiadis et al., 2018; Jennings

et al., 2019; Dalton et al., 2020). Recently, Jakobsen et al. (2018) and Reusche et al. (2018) proposed overall glacial retreat of the GrIS in north Greenland during the Holocene, but with a possible stillstand of the ice sheet and its outlet glaciers as a response to the 8.2 cold events. These studies further suggested a restricted extent of the ice sheet in middle and late Holocene, until c. 0.3 ka where the ice reached its Little Ice Age (LIA) position.

The glacial history of Inglefield Land comprises the history of the Smith Sound Ice Stream along the coast, and the history of

the GrIS in Inglefield Land, as described by Nichols (1969), Tedrow (1970) and Blake et al. (1992), with additional evidence from the neighbouring Humboldt Glacier and Washington Land to the east by Bennike (2002) and Reusche et al. (2018). The deglaciation of the interior parts of Inglefield Land is less well known but a set of distinct moraine systems between the present-day ice margin and the coast line (Nichols, 1969) suggest that the GrIS made several stops or readvances during the overall deglaciation of Inglefield Land. Our current knowledge about the timing of deglaciation in Inglefield Land comprises a number

of minimum limiting radiocarbon ages of raised marine deposits from the coastal areas that range from c. 8.6 to 6.6 cal. ka BP (Nichols, 1969; Blake et al., 1992; Mason, 2010). The marine limit in Inglefield Land has been determined at several locations



and decreases from c. 90 m in the southwestern part of Inglefield Land to c. 65 m in the northeastern part of Inglefield Land (Nichols, 1969; Funder and Hansen, 1996).

## 3 Methods

### 3.1 Cosmogenic nuclide exposure dating

Cosmogenic nuclide exposure dating is a widely used method to constrain the deglaciation history of former glaciated areas (Gosse and Phillips, 2001; Ivy-Ochs and Kober, 2008; Balco, 2020). One of the most common used nuclides is [10]Be as it forms in the abundant mineral quartz, and is fairly easy to extract and measure by accelerator mass spectrometry. However, due to its relatively slow decay and long half-life ($1.4 \times 10^6$ yr), problems can arise in areas characterized by slow-moving cold-based ice as small rates of erosion hinder complete removal of nuclides "inherited" from prior exposures and thus, yield apparent exposure ages exceeding the length pf the last ice-free period (Heyman et al., 2011). Nuclide inheritance is present in samples throughout Greenland, particularly at high elevations away from glacial troughs and fjords (Kelly et al., 2008; Corbett et al., 2013; Håkansson et al., 2016; Young et al., 2020) but it seems to be especially frequent in north Greenland where several studies have shown more widespread nuclide inheritance (Farnsworth et al., 2018; Larsen et al., 2018; Søndergaard et al., 2019; Ceperley et al., 2020; Larsen et al., in review).

Measurements of in-situ produced [14]C in boulders and bedrock can however circumvent the nuclide inheritance problem and help to obtain more reliable exposure ages (Hippe, 2017; Graham et al., 2019). Due to its shorter half-life (5730 yr), in-situ [14]C is sensitive to radioactive decay on late Quaternary and Holocene timescales as the concentration build up from prior exposure will rapidly decrease, when a surface is shielded from cosmic rays (Lifton et al., 2001; Hippe, 2017). As such, it is an optimal tool to solve the most recent deglaciation history of the GrIS. However, in-situ [14]C is still not the preferred nuclide for exposure dating as the extraction process is demanding despite many improvements and developments within recent years (Lifton et al., 2015; Goehring et al., 2019; Lupker et al., 2019). Still more robust information on the deglaciation history can be achieved by using combined measurements of [10]Be and [14]C as shown by previous studies (Corbett et al., 2013; Hippe, 2017; Young et al., 2018; Graham et al., 2019).

### 3.1.1 [10]Be exposure dating

[10]Be exposure dating of boulders and pebbles was used to constrain the most recent glacial history of Inglefield Land. A total of 25 boulder samples were collected, all resting on bedrock except for sample GL1732-GL1735, which were on top of two moraines in the western part of Inglefield Land (Fig. 1c). In addition, two samples consisting of quartz pebbles (GL1715 and GL1716) were collected on an outwash plain in the northeastern part of Inglefield Land. Samples were collected using a rock saw, hammer and chisel to cut out the top few centimetres of quartz bearing stable boulders (Fig. 2). With a hand-held Garmin e-trex 30 GPS, we recorded the latitude, longitude and elevation of each sample. The orientation of the rock surface and



shielding by the surrounding topography were measured using a compass and clinometer, respectively. Elevations of the sampled boulders were all between 65 m a.s.l. and 542 m a.s.l., and thus, were all at or above the local marine limit of the area

(Nichols, 1969; Blake et al., 1992; Funder and Hansen, 1996). We measured sample thicknesses with a caliper before the samples were crushed and sieved. This information was used to calculate the average thickness of each sample. For boulder and pebble samples we used the 250-700 μm fraction to isolate quartz and extract beryllium.

All samples were processed in the Cosmogenic Nuclide Laboratory at Department of Geoscience, Aarhus University following methods adapted from (Corbett et al., 2016). The $^{10}$Be/$^9$Be ratios were measured at the Aarhus AMS Centre and all samples

were blank corrected. Nuclide concentrations were normalized to the Beryllium standard ICN-01-5-4, with a $^{10}$Be/$^9$Be value of 2.851 x 10$^{-12}$ (Nishiizumi et al., 2007). Apparent $^{10}$Be exposure ages were calculated using the online exposure age calculator formerly known as the CRONUS-Earth online exposure calculator v.3 (Balco et al., 2008) in combination with the Baffin Bay production rate (Young et al., 2013) and the Lm production scaling scheme (Lal, 1991; Stone, 2000). The rock density was set to 2.65 g/m$^3$ as it is representative for the boulders we sampled, and we assumed zero erosion. We did not correct for cover by

vegetation or snow as the vegetation in the area is sparse and precipitation rates are low, c. 100-150 mm per year (Dawes, 2004). The sampled boulders were furthermore all positioned in open locations in the landscape making it highly unlikely that any snow cover persisted for long periods of time. As the glaciostatic uplift history is not well constrained in north Greenland, we present our $^{10}$Be ages without any uplift correction, similarly to many other studies in Greenland which have shown the corrections to be negligible (Young et al., 2012; Sinclair et al., 2016; Larsen et al., 2018; Young et al., 2020).

All resulting apparent exposure ages and parameters used in the calculations can be seen in Table 1. The $^{10}$Be ages are presented with 1σ analytical uncertainty and ages calculated using other scaling schemes deviate by <2 %.

### 3.1.2 In-situ $^{14}$C exposure dating

We used in-situ $^{14}$C exposure dating to further constrain the deglaciation history of Inglefield Land, primarily by testing for

$^{10}$Be nuclide inheritance in selected samples. Four of the quartz samples used for $^{10}$Be exposure dating were chosen for in-situ $^{14}$C exposure dating, GL1725, GL1712, GL1701 and GL1708. We chose these samples based on i) the resulting apparent $^{10}$Be exposure ages within each sample location, ii) the amount of quartz left, and iii) the sample location in the study area to secure a broad spatial distribution (Fig. 1c). Approximately 4 g of purified quartz, the same used for $^{10}$Be extraction, was used to extract the in-situ produced $^{14}$C. Samples for in-situ $^{14}$C measurements were processed using the in-situ $^{14}$C extraction line at

ETH Zürich (Hippe et al., 2009; Lupker et al., 2019). Samples were measured at ETH Zürich with the MICASAS AMS system (Synal et al., 2007; Wacker et al., 2010) and sample in-situ $^{14}$C concentrations were calculated from measured $^{14}$C/$^{12}$C ratios (Hippe and Lifton, 2016). In-situ $^{14}$C ages were calculated using the online exposure age calculator formerly known as the CRONUS-Earth online exposure calculator v.3 (Balco et al., 2008), the west Greenland production rate (Young et al., 2014), and the Lm production scaling scheme (Lal, 1991; Stone, 2000). All resulting ages and variables used in the calculations are



listed in Table 2. In-situ [14]C exposure ages are presented with 1σ analytical uncertainty and ages calculated using other scaling

schemes deviate by <4 %.

## 3.2 Radiocarbon dating of reworked molluscs and wood fragments

Radiocarbon dating of reworked organic material in glacial deposits can be used to determine when the ice extent was smaller

than present (Bennike and Weidick, 2001; Briner et al., 2014; Farnsworth et al., 2018). For this purpose, we therefore collected

reworked marine molluscs at the southern margin of the Humboldt Glacier (Fig. 1c). From the samples site, 15 molluscs were

chosen, pre-treated following the procedure of (Brock et al., 2010), and radiocarbon dated at the Aarhus AMS Centre (Olsen

et al., 2016). In addition, four wood fragments were retrieved at 193 m a.s.l. on the meltwater plain in front of the Hiawatha

Glacier (Fig. 1c). The wood fragments were pre-treated and radiocarbon dated at Beta Analytic.

Radiocarbon ages for the molluscs were calibrated using OxCal v4.3 (Ramsey, 2009) with the Marine13 calibration curve

(Reimer et al., 2013) and a marine reservoir effect of 550 [14]C years (ΔR=150 [14]C a) based on a couple of ages from molluscs

collected alive before 1960 in north Greenland (Mörner and Funder, 1990). Radiocarbon ages for the wood fragments were

calibrated with the IntCal13 calibration curve (Reimer et al., 2013). Sample information, resulting radiocarbon ages, and

calibrated ages are reported in Table 3. Throughout the text, we use the mean calibrated radiocarbon age ±2σ.

**4 Results**

### 4.1 [10]Be and in-situ [14]C exposure dating

[10]Be exposure dating was carried out on 25 boulder samples and 2 samples consisting of pebbles to constrain the deglaciation

of Inglefield Land (Fig. 3). The measured [10]Be concentrations in the 27 samples range from $3.0\pm0.9\times10^4$ to $60.3\pm0.9\times10^4$ [10]Be

at/g and result in apparent exposure ages ranging from 92.7±1.5 ka to 6.8±2.0 ka, with the oldest ages being from boulders on

moraines in the western part of the area and the younger ages resulting from boulders closer to Humboldt Glacier and the coast

in the northeastern part of the area (Fig. 3, Table 1).

Although the [10]Be ages are scattered we see some structure in the dataset. The majority of ages sampled below 300 m a.s.l.

group within the post-LGM period, with a peak in early Holocene whereas most samples above 450 m a.s.l. predate the LGM.

Further, there also seem to be a vague pattern in spatial distribution, with the oldest [10]Be ages being from the two moraine

ridges in western Inglefield Land and the youngest boulder ages closer to Humboldt Glacier.

In-situ [14]C exposure dating were carried out to better constrain the deglaciation of Inglefield Land from the scattered [10]Be ages.

The measured in-situ [14]C concentrations in the four samples range from $8.4\pm0.3\times10^4$ to $17.4\pm0.2\times10^4$ at/g and resulted in

exposure ages ranging from 14.2±0.5 ka to 6.7±0.3 ka (Fig. 3, Table 2). All in-situ [14]C ages are younger than the [10]Be ages

resulting from the same quartz sample, confirming that the [10]Be ages are generally affected by nuclide inheritance. However,



the in-situ [14]C ages do to some degree match the youngest [10]Be ages from the same localities, except for GL1701, which predate the Holocene. The remaining three in-situ [14]C ages group in middle Holocene.

### 4.2 Radiocarbon dating of reworked molluscs and wood fragments

Reworked marine molluscs were collected from the surface of and within diamictic sediments along the western margin of the
Humboldt Glacier (Fig. 1c). Several species were identified and 15 samples of *Mya truncata*, *Hiatella arctica* and *Astarte borealis* were used for radiocarbon dating. The calibrated mean radiocarbon ages range from 3.6±0.04 to 0.5±0.03 cal. ka BP (Fig. 4, Table 3) and reflect the period when the Humboldt Glacier was behind its present-day extent. In addition, the wood samples collected on the outwash plain in front of the Hiawatha Glacier resulted in ages between 5.8±0.06 cal. ka BP and 1.9±0.04 cal. ka BP (Fig. 4, Table 3).

# 5 Discussion

### 5.1 Indications of low erosion rates and cold-based ice in north Greenland

The [10]Be ages from Inglefield Land are scattered which makes it difficult to fully constrain the glacial history in the area. We consider the 12 [10]Be ages older than the LGM as evidence of nuclide inheritance from prior exposure (Fig. 3) and discard them as constraints of the glacial history in Inglefield Land. Of the remaining 15 samples postdating the LGM, some do most likely
also reflect nuclide inheritance. Within uncertainty, only four of the 15 post-LGM [10]Be ages overlap with the three youngest in-situ [14]C ages. Thus, by including in-situ [14]C ages in the analysis we determine that 11 of the post-LGM [10]Be ages are affected by inheritance and thus, overestimate the post-LGM exposure period to a varying degree. In total, 24 out of 27 samples (c. 89 %) from Inglefield Land show some degree of nuclide inheritance.

We also consider the oldest in-situ [14]C age of c. 14.2 ka to be affected by inheritance as it is unlikely that Inglefield Land was
deglaciated at that time. The only modelled scenario of nuclide build up that almost reach the measured concentration of the sample and still follow the known glacial history of the GrIS in north Greenland is seen in Figure 5. In this scenario, Inglefield Land was deglaciated during MIS 3 from 45 to 23 ka and again in Holocene from 6.7 ka until present. This limits the expansion of the GrIS during the LGM to a narrow interval from c. 23 to 7 ka. This scenario is to some degree consistent with other studies in northern Greenland that suggest a restricted GrIS during MIS 3 (Larsen et al., 2018; Søndergaard et al., 2019) and a
late coalescence of the GrIS and Inuitian Ice Sheet around 22 cal. ka BP (England, 1999). However, as we only have one datapoint and the simulation is incapable of fully reaching the measured concentration we cannot make any firm conclusions on the timing of prior exposure of the sample and the implications for the ice sheet history.

The trend between apparent [10]Be ages and elevation of the samples points towards larger amount of inheritance in samples from higher elevations (Fig. 6). This pattern has also recently been observed in adjacent Washington Land as well as in Dove
Bugt, northeast Greenland (Ceperley et al., 2020; Skov et al., 2020). In addition, there is an increasing amount of inheritance





in samples farther away from the Humboldt Glacier. This spatial distribution of samples with inheritance at higher elevations away from the Humboldt Glacier is expected as these locations represent areas outside troughs where erosion is low because of slowly moving or even cold-based ice. A similar relationship between nuclide inheritance and elevation and distance to deep fjords with large fast flowing outlet glaciers indicative of higher erosion rates has been demonstrated elsewhere in

Greenland (Larsen et al., 2014; Søndergaard et al., 2019).

Overall, inheritance and the lack of sufficient nuclide resetting is a widespread problem especially in north Greenland, and have complicated several studies within recent years (Corbett et al., 2015; Farnsworth et al., 2018; Søndergaard et al., 2019; Ceperley et al., 2020; Larsen et al., in review). Thus, we conclude that large parts of the north GrIS were inefficient at eroding the subglacial topography during parts of or throughout the last glaciation probably because subglacial sliding were limited by

cold-based thermal conditions and the overall low ice flux resulting from the relatively small precipitation rates of the region. We note that cold-based zones are also considered to dominate the present-day thermal state of the GrIS (MacGregor et al., 2016).

## 5.2 Holocene glacial history of Inglefield Land

During the LGM, Inglefield Land was completely ice covered and the ice nourished the Smith Sound Ice Stream primarily through the Humboldt Glacier until the opening of Nares Strait sometime between c. 9 and 8 cal. ka BP (Georgiadis et al., 2018; Jennings et al., 2019; Dalton et al., 2020). The outer coast at Kap Inglefield Land, Kap Grinell and Renselaer Bay in southwest Inglefield Land was deglaciated between c. 8.6 and 8.3 cal. ka BP (Nichols, 1969; Blake et al., 1992; Mason, 2010). Farther north, the deglaciation of the outer coast at Marshall Bay in central Inglefield Land is constrained to c. 7.9 ka by a

single in-situ $^{14}$C age. This age is largely consistent with a radiocarbon age of marine molluscs of c. 8.2 cal. ka BP at Minturn Elv located c. 20 km east of Marshall Bay (Nichols, 1969) (Fig. 7a).

After reaching the outer coast, the ice margin continued its retreat towards its present-day position which was reached by c. 6.7 ka in the central part of Inglefield Land. Farther north, the ice probably retreated somewhat slower as suggested by dating of a rearrangement of the meltwater drainage pattern from the Hiawatha Glacier. Initially, meltwater flowed from the Hiawatha

Glacier towards Dallas Bay, but this changed when the ice margin was approximately halfway between the coast and present-day extent where a water divide then rerouted the meltwater towards Marshall Bay (Nichols, 1969). The timing of this change is constrained by a single $^{10}$Be age of meltwater deposits (pebble sample) that yield an age of c. 6.8 ka (Fig. 7b). The $^{10}$Be age is, however, consistent with a radiocarbon age of molluscs, presumably from lower lying prodeltaic sediments, which gave an age of c. 6.6 cal. ka BP at Dallas Bay (Nichols, 1969). This suggests that the ice margin was located north of the meltwater

drainage divide around c. 6.8 ka. Farthest north in Inglefield Land, at the southern flank of the Humboldt Glacier the ice margin reached its present-day extent already by c. 8.2 ka (Fig. 7a). This age is consistent with the $^{10}$Be chronology from the northern flank of Humboldt Glacier where a moraine a few hundred meters outside the LIA moraine was abandoned c. 8.3 ka (Reusche et al., 2018).





After the ice margin reached its present-day position it continued to retreat farther inland. Wood fragments in front of the
Hiawatha Glacier demonstrate that the land-based part of the GrIS in Inglefield Land was smaller than present between c. 5.8
and 1.9 cal. ka BP. In addition, the age distribution of the molluscs collected at the Humboldt Glacier margin indicates that it
was behind its present-day position between c. 3.6 to 0.5 cal. ka BP (Fig. 7a). At the northern flank of Humboldt Glacier,
radiocarbon ages of reworked marine molluscs suggest that the glacier retreated at least 25 km farther inland between c. 3.7
and 0.3 cal. ka BP (Bennike, 2002) possibly favoured by the bed topography being below sea level 10's of km inland
(Morlighem et al., 2014; Morlighem et al., 2017). Thus, no later than c. 0.3 cal. ka BP, the GrIS in Inglefield Land re-advanced
towards its LIA maximum extent. The spatial extent of the late Holocene retreat of the Hiawatha Glacier behind its present-
day ice margin is not known, but the ice retreat may possibly have exposed parts of the Hiawatha impact crater (Fig. 7b).

**5.3 A review of the Holocene glacial history in northwest and north Greenland**

In the following we review the new data from north and northwest Greenland to put our results into a broader context. We
focus on two stages of the deglaciation history of the GrIS, namely when it i) deglaciated from the coast towards its present-
day extent and ii) when it was smaller than present (Fig. 8). For information about the offshore deglaciation history, the reader
is referred to recent reviews by Georgiadis et al. (2018), Jennings et al. (2019) and Dalton et al. (2020).

In the southern part of northwest Greenland, the ice margin reached the outer coast near Upernavik c. 11.3 ka (Corbett et al.,
2013) coinciding with the overall ice retreat in Melville Bay initiating c. 11.6 ka (Søndergaard et al., in review) (Fig. 8b). The
ice reached the inner part of Upernavik Fjord c. 9.9 ka (Briner et al., 2013) and in Melville Bay the ice was at its present-day
extent already c. 11.5 ka (Søndergaard et al., in review) (Fig. 8b). Farther north, coastal deglaciation near Thule and Delta Sø
began c. 10.8 ka (Corbett et al., 2015; Axford et al., 2019) and the ice margin reached its present-day extent at Delta Sø c. 10.1
ka (Axford et al., 2019). In Inglefield Bredning north of Thule the ice reached the inner parts of the fjord c. 11.9 ka
(Søndergaard et al., 2019) (Fig. 8b). In north Greenland, results from Inglefield Land show that the deglaciation of the outer
coast commenced c. 8.6 cal. ka BP in southeast and c. 7.9 ka in the central part of the coast line and the ice reached its present-
day position in central Inglefield Land c. 6.7 ka. The Humboldt Glacier deglaciated and reached its present-day extent c. 8.2
ka. In the adjacent Washington Land, deglaciation of the outer coast is constrained to c. 9.0 ka with widespread deglaciation
of the entire area evident c. 8.6 ka (Ceperley et al., 2020). Farther north, Petermann Glacier was positioned at the outer sill in
Hall Basin c. 8.7 (Jakobsson et al., 2018) and it reached its present day position c. 6.9 ka (Reilly et al., 2019).

Restricted ice extent behind the present-day ice margin in northwest and north Greenland was widespread during large parts
of middle and late Holocene. In Upernavik and Melville Bay, the ice sheet was behind its present-day position between c. 9.1
and 0.4 cal. ka BP (Bennike, 2008; Briner et al., 2013; Briner et al., 2014; Axford et al., 2019; Søndergaard et al., in review)
(Fig. 8c). Farther north in the Thule area and around Qaanaaq, mosses from a local ice cap and subfossil plants from the GrIS
show a smaller ice extent before c. 3.3 cal. ka BP (Farnsworth et al., 2018; Axford et al., 2019; Søndergaard et al., 2019). In
Inglefield Land, north Greenland, wood fragments in front of the Hiawatha Glacier suggest that the ice margin was behind its



present-day extent from c. 5.8 and 1.9 cal. ka BP, whereas the Humboldt Glacier retreated at least 25 km inland c. 3.7 to 0.3 cal. ka BP (Bennike, 2002). The Petermann Glacier farther north also retreated farther inland of its present-day position and following readvance reached its LIA extent c. 0.3 ka (Reusche et al., 2018; Reilly et al., 2019) (Fig. 8c).

In summary, the timing of deglaciation along the coast in northwest Greenland is earlier than in north Greenland around Nares Strait, but the timing is, however, in accordance with the overall deglaciation in Greenland (Bennike and Björck, 2002; Funder et al., 2011). Further, the periods of middle and late Holocene restricted ice extent of the larger outlet glaciers in north Greenland initiated later than in northwest Greenland and was shorter.

## 5.4 Holocene ice and climate interactions in northwest and north Greenland

The contrasting pattern of deglaciation between northwest and north Greenland, can in part be explained by different responses of the two sectors to Holocene climate changes (Fig. 9). The early deglaciation of the land areas in northwest Greenland from Upernavik to Inglefield Bredning coincides with the early Holocene Thermal Maximum (HTM) (c. 11-8 ka) in northwest Greenland (Lecavalier et al., 2017; Axford et al., 2019) which initiated earlier than in the rest of Greenland (Briner et al., 2016;

Buizert et al., 2018) (Fig. 9a-c). This rapid increase in surface air temperatures has been suggested to be the main driver of the widespread rapid deglaciation specifically in Melville Bay, the Thule area and Inglefield Bredning (Axford et al., 2019; Søndergaard et al., 2019; Søndergaard et al., in review), showing the sensitivity of marine based ice to rising air temperatures. Although it has been suggested that warm Atlantic waters in Hall Basin, northern Nares Strait, assisted early Holocene ice retreat (Jennings et al., 2011), warm waters and increasing ocean temperatures in southern Nares Strait seemed to arrive later

than along the west Greenland coast (Dyke et al., 1996; Levac et al., 2001; Lecavalier et al., 2017; Axford et al., 2019). This delay in warming ocean conditions in southern Nares Strait might be the reason why the opening of Nares Strait and the deglaciation of the coastal areas in Inglefield Land and Washington Land happened 2-3 ka later than the land areas in northwest Greenland (Bennike and Björck, 2002; Larsen et al., 2014; Sinclair et al., 2016; Larsen et al., 2019).

After the ice margin reached its present-day extent in north and northwest Greenland it continued to retreat farther inland. In

northwest Greenland, the period with restricted ice extent in Melville Bay c. 9.1 to 0.4 cal. ka BP, was driven by a strengthening of the West Greenland Current and warm ocean waters arriving in middle Holocene (Levac et al., 2001; Caron et al., 2020) (Fig. 9d). The presence of *Chlamys islandica* infers that the period with marine based outlets were behind their present-day extent in North Greenland coincides with the arrival of warm waters in Nares Strait (Bennike, 2002). The Neoglacial cooling are known to have affected ice on land in northwest Greenland and resulting in expansion of local ice caps, lake ice cover and

even parts of the northwest GrIS (Blake et al., 1992; Lasher et al., 2017; Farnsworth et al., 2018; Søndergaard et al., 2019). The Hiawatha Glacier in north Greenland show a re-advance after c. 1.9 cal. ka BP, as a possible response to the Neoglacial cooling, which also seems to have provoked a readvance of the Petermann Glacier c. 2.8 ka (Reusche et al., 2018). Finally, the ice in north and northwest Greenland show a near synchronous re-advance towards its LIA extents which coincides with the LIA cooling within the last millennium (Lasher et al., 2017; Lecavalier et al., 2017; Axford et al., 2019).




## 6 Conclusion

In this study we used in-situ [10]Be and [14]C cosmogenic nuclide exposure dating and radiocarbon dating of reworked organic material to constrain the Holocene glacial history of Inglefield Land, north Greenland. Our results revealed a large scatter in the [10]Be ages with c. 45 % of the ages predating the LGM and an overall 89 % of the samples being affected by inheritance

possibly due to low-erosive cold-based ice. We find that the outer coast in Inglefield Land began to deglaciate between c. 8.6 and 8.3 cal. ka BP in the southeastern part, whereas the central part was deglaciated by c. 7.9 ka. Following initial deglaciation, the ice margin reached its present-day position c. 6.7 ka in central Inglefield Land, whereas Humboldt Glacier in the northern part of the study area reached its present extent already by c. 8.2 ka. After deglaciation the ice margin retreated behind its present-day extent from c. 5.8 to 1.9 cal. ka BP at the Hiawatha Glacier and c. 3.7 to 0.3 cal. ka BP at the Humboldt Glacier.

Thus, the readvance towards the LIA extent initiated between 1.9 and 0.3 cal. ka BP.

We furthermore reviewed new data from north and northwest Greenland to put our results into a broader context and assessed the findings with local and regional climate records. We find that the Holocene glacial history varies significantly between northwest and north Greenland. The deglaciation from the coast to the present-day ice extent in northwest Greenland occurred at the onset of the Holocene, possibly as a response to the relatively early HTM. Deglaciation continued and the ice sheet

retreated behind its present-day extent in northwest Greenland throughout most of middle and late Holocene driven by continued high air temperatures and the arrival of warm waters along the west Greenland coast. Contrary, the deglaciation of the outer coast in Nares Strait and north Greenland was delayed c. 2-3 ka and show a more restricted period of retreat behind its present-day extent. The observed difference in pattern of deglaciation in the two regions is most likely a consequence of the large marine based part of the northwest GrIS being more sensitive to climate changes as opposed to the largely land based

north GrIS. Further, the late opening of Nares Strait could have delayed ice retreat in north Greenland, despite early atmospheric warming. During the LIA cooling, the GrIS do though show a synchronous response with ice advance throughout north and northwest Greenland. Our findings highlight the complexity of the ice-climate system and show clear differences in ice sheet sensitivity between northwest and north Greenland. As such, this add new knowledge and possible constraints on the future state of the GrIS as a response to present-day global warming.

**Author contribution**

N.K.L, K.H.K, S.F and A.S.S participated in fieldwork and decided on the sampling strategy. A.S.S did [10]Be sample preparation and J.O. and A.S.S carried out measurements and calculation. O.S and A.S.S carried out in-situ [14]C sample preparation, measurements and calculation. A.S.S and N.K.L made initial interpretations of the results and wrote the paper with contribution from the co-authors.



**Acknowledgement**

This research was supported by Aarhus University Research Foundation and the Villum Foundation. Birte Lindahl Eriksen and Rikke Brok Jensen are thanked for extensive help in the laboratory. We also thank the Carlsberg Foundation for supporting this study.

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



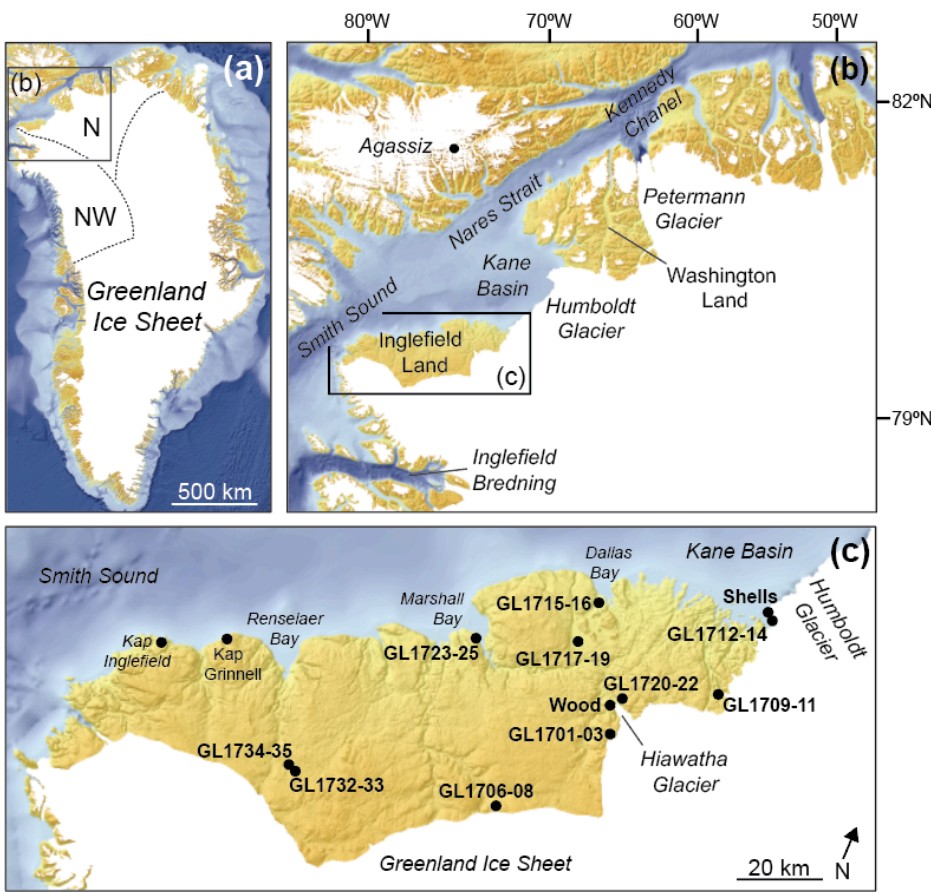

**Figure 1:** Location of the study area in Greenland (a), with marked northwest (NW) and north (N) Greenland extent and places mentioned in the text (b). (c) shows Inglefield Land with places discussed in the text. Black dots denote sample locations for wood fragments in front of the Hiawatha Glacier, molluscs at the margin of the Humboldt Glacier and boulder samples (GL17XX) collected throughout the study area.



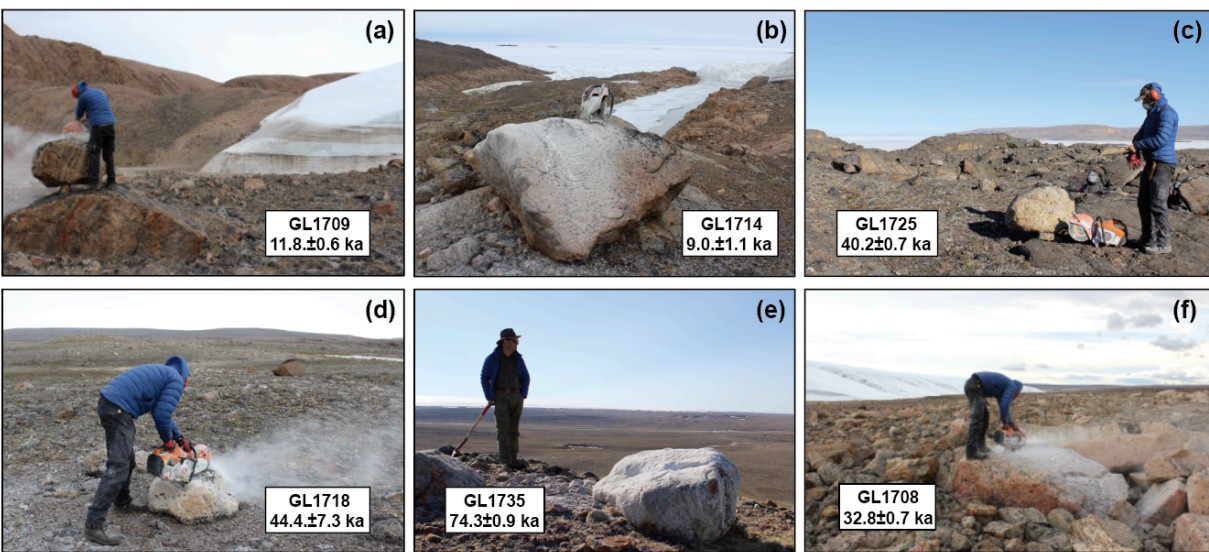

**Figure 2:** Boulders sampled for [10]Be exposure dating in Inglefield Land ((a)-(f)), as well as resulting ages. (a), (f) show boulders sampled close to the present-day ice margin. (b) is a boulder sampled next to the Humboldt Glacier. (c), (d) show boulders samples close to the outer coast. (e) shows a boulder sampled on a moraine in the western part of the study area.





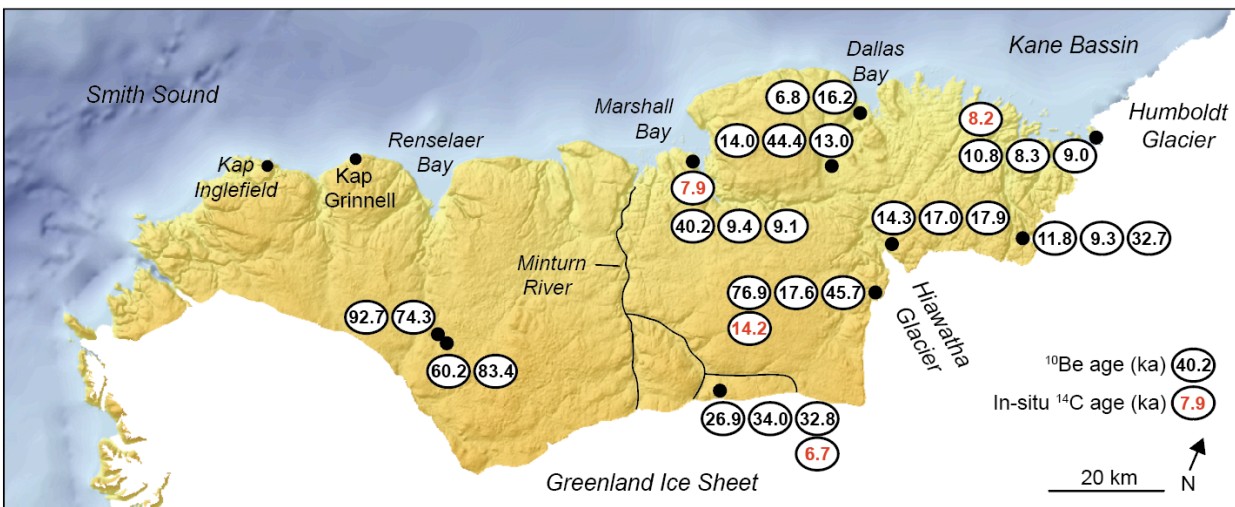

**Figure 3:** Map of Inglefield Land, north Greenland. Black dots denote sample locations for boulder samples and their resulting ¹⁰Be and in-situ ¹⁴C exposure ages given in ka.




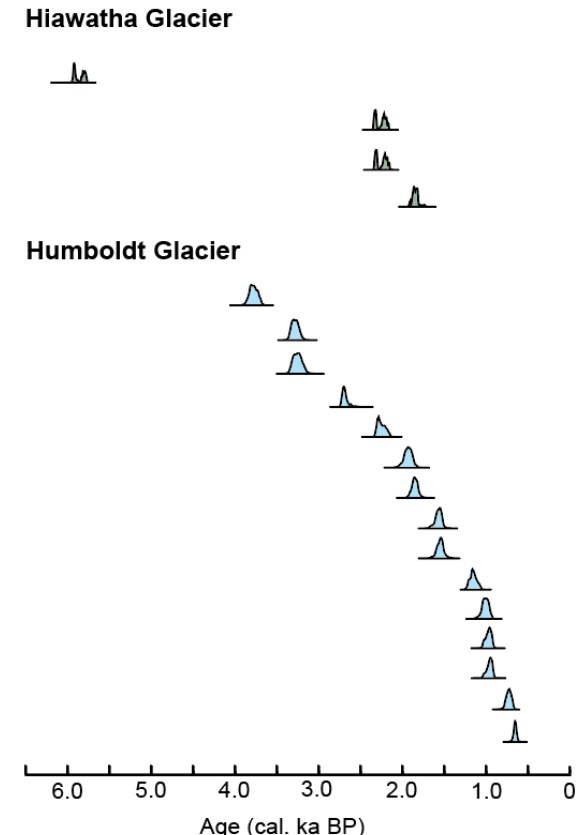

**Figure 4:** Radiocarbon age probability plots of wood fragments collected in front of the Hiawatha Glacier (green) and reworked marine molluscs collected at the margin of Humboldt Glacier (blue). Each plot represents the age of a single wood fragment or mollusc shell and its calibrated age probability distribution.




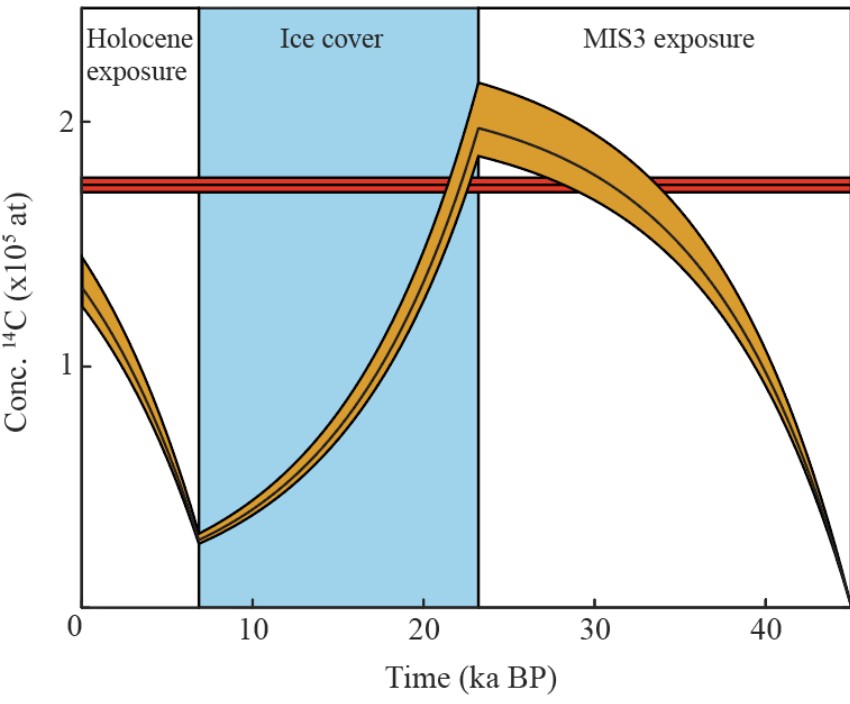

**Figure 5:** Modelled exposure scenario for boulder sample GL1701. The red line shows the measured in-situ $^{14}$C concentration in the sample
and the orange line is the nuclide concentration build up and decay during periods of exposure from 45 to 23 ka and again from 6.7 until
present and ice cover in between.





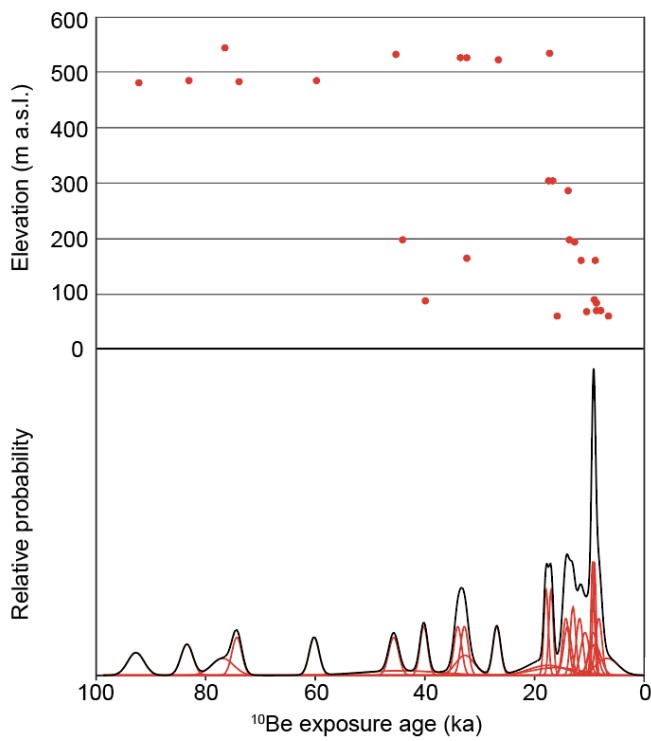

**Figure 6:** [10]Be ages from boulders sampled at various elevations in Inglefield Land, north Greenland. The top panel shows [10]Be ages plotted against elevation of the sample sites. Each red dot represents the [10]Be age of an individual boulder and its associated elevation. The bottom panel shows the relative probability distribution of the boulder ages with their 1σ analytical uncertainty (red lines) and the cumulated probability plot of all [10]Be ages (black line).






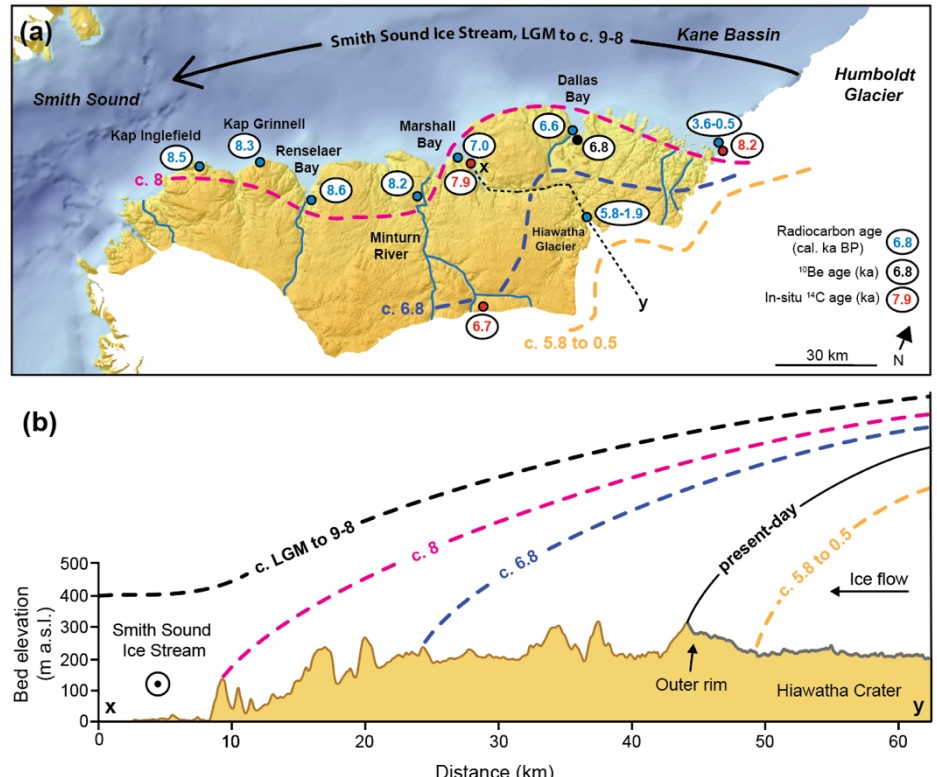

**Figure 7:** Deglaciation in Inglefield Land. (a) shows ages believed to constrain the deglaciation in the area. In-situ $^{14}$C ages and radiocarbon ages at Humboldt Glacier and Hiawatha Glacier are from this study. Radiocarbon ages at the coast are from Nichols (1969), Blake et al. (1992) and Mason (2010). (b) shows the Holocene deglaciation pattern in Inglefield Land inferred from this study across the transect (x-y) seen in (a).






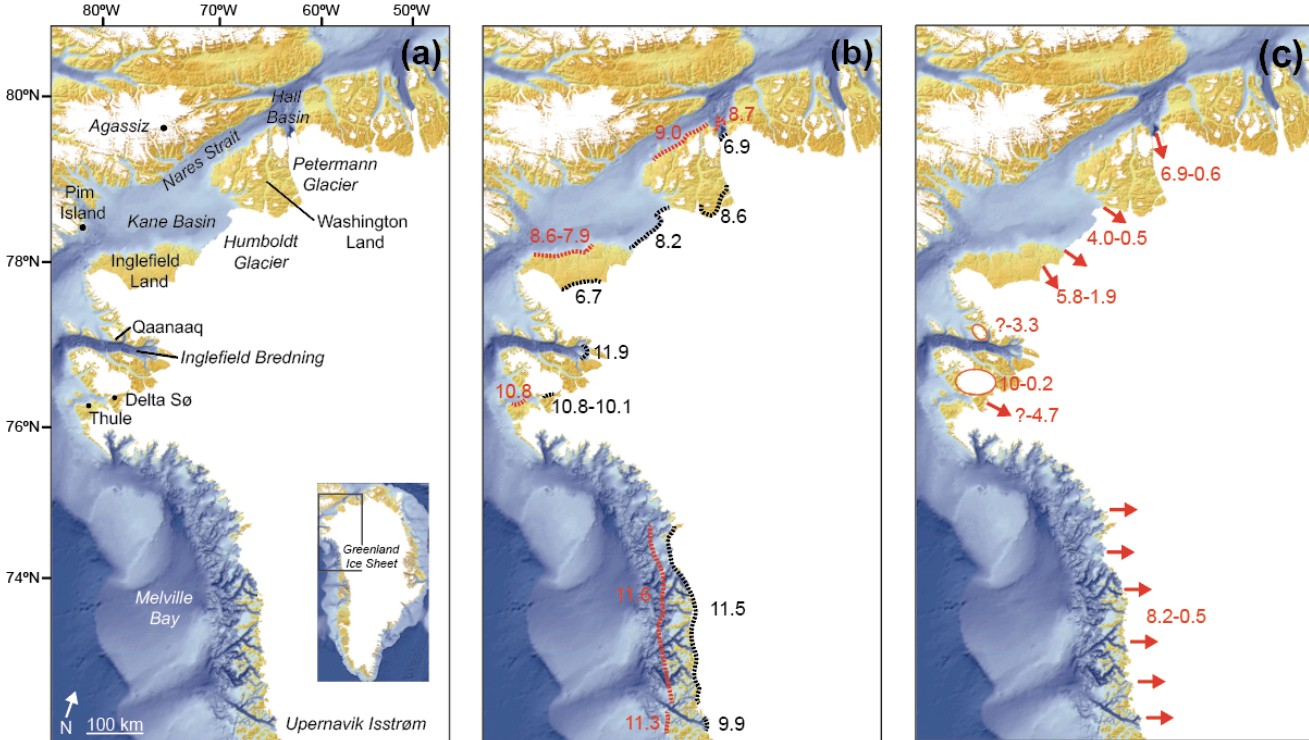

**Figure 8**: Age constraints (in ka) of the Holocene ice extent at the outer coast, present-day ice margin and for the period when the GrIS was
smaller than present. (a) shows localities in northwest and north Greenland. (b) shows the initial Holocene deglaciation of the GrIS towards
the outer coast (red) (Corbett et al., 2013; Corbett et al., 2015; Jakobsson et al., 2018; Ceperley et al., 2020; Søndergaard et al., in review)
and following retreat towards the present-day ice margin (black) (Briner et al., 2013; Reusche et al., 2018; Axford et al., 2019; Reilly et al.,
2019; Søndergaard et al., 2019; Ceperley et al., 2020; Søndergaard et al., in review). (c) shows the period when the GrIS and its outlets
(arrows) and local ice caps (circles) were smaller than the present-day extent (Bennike, 2002; Bennike, 2008; Briner et al., 2014; Farnsworth
et al., 2018; Axford et al., 2019; Reilly et al., 2019; Søndergaard et al., 2019; Søndergaard et al., in review). A question mark means that the
upper limit of restricted ice extent has not been constrained.

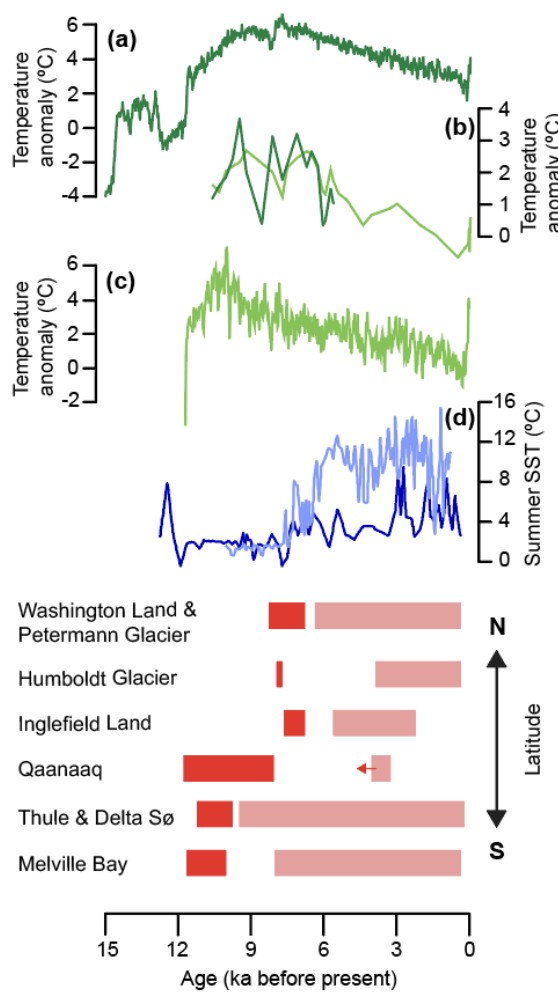

**Figure 9:** Ice extent and climate fluctuations for north and northwest Greenland during the last 15 ka. (a) mean annual temperature anomalies, northwest Greenland (Buizert et al., 2018). (b) July air temperature anomalies at Delta Sø, northwest Greenland, using two different training sets and transfer functions, FOR15 (light green) and FRA06 (dark green) (Axford et al., 2019). (c) Agassiz $\delta^{18}O$ temperature reconstruction (Lecavalier et al., 2017). (d) Reconstructed west Greenland sea surface temperatures from Gibb et al. (2015) (dark blue) and Ouellet-Bernier et al. (2014) (light blue). The lower panel shows the inferred ice extent in north and northwest Greenland from Melville Bay in the south to Washington Land in the north (for references, see text). Dark red bars denote known periods of deglaciation from the outer coast towards the present-day ice margin and bright red bars denote periods of smaller than present-day extent. Arrows indicate the lack of an upper or lower constraint.




**Table 1:** Sample collection, [10]Be isotopic information and resulting exposure ages for 25 boulder and two pebble samples from Inglefield Land, north Greenland.

| Sample name | Sample type | Elevation (m a.s.l.) | Latitude (°N) | Longitude (°W) | Sample thickness (cm) | Shielding correction | Quartz (g) | Carrier added (g)[a] | Sample [10]Be/[9]Be ratio (10[-14]) | Blank [10]Be/[9]Be ratio (10[-14]) | [10]Be conc. (atoms/g) | [10]Be unc. (atoms/g) | [10]Be age (ka)[b] |
|---|---|---|---|---|---|---|---|---|---|---|---|---|---|
| GL1701 | Boulder | 542 | 78.76 | -67.01 | 5.8 | 0.9999 | 40.006 | 0.739 | 130.9±2.8 | 0.3±0.08 | 529068 | 13417 | 76.9±2.0 |
| GL1702 | Boulder | 533 | 78.76 | -67.01 | 5.2 | 0.9997 | 40.046 | 0.751 | 30.0±5.7 | 0.3±0.08 | 122306 | 23513 | 17.6±3.4 |
| GL1703 | Boulder | 530 | 78.76 | -67.02 | 5.7 | 0.9994 | 40.084 | 0.762 | 75.4±1.1 | 0.3±0.08 | 312981 | 6183 | 45.7±0.9 |
| GL1706 | Boulder | 521 | 78.51 | -67.94 | 5.3 | 0.9997 | 40.176 | 0.751 | 45.3±1.0 | 0.3±0.08 | 184398 | 4623 | 26.9±0.7 |
| GL1707 | Boulder | 524 | 78.51 | -67.94 | 5.7 | 0.9997 | 40.428 | 0.756 | 57.0±0.9 | 0.3±0.08 | 232494 | 4811 | 34.0±0.7 |
| GL1708 | Boulder | 524 | 78.51 | -67.94 | 5.4 | 0.9998 | 40.071 | 0.752 | 55.0±1.0 | 0.3±0.08 | 225228 | 5069 | 32.8±0.7 |
| GL1709 | Boulder | 159 | 78.94 | -67.02 | 5.4 | 0.9952 | 40.043 | 0.602 | 17.2±0.8 | 0.3±0.08 | 55336 | 2595 | 11.8±0.6 |
| GL1710 | Boulder | 159 | 78.94 | -67.02 | 5.3 | 0.9963 | 40.050 | 0.603 | 13.7±1.1 | 0.3±0.08 | 43909 | 3758 | 9.3±0.8 |
| GL1711 | Boulder | 163 | 78.94 | -66.02 | 4.8 | 0.9950 | 40.065 | 0.205 | 47.6±2.5 | 0.02±0.01 | 154443 | 8090 | 32.7±1.7 |
| GL1712 | Boulder | 65 | 79.14 | -65.81 | 5.6 | 0.9998 | 34.035 | 0.206 | 11.9±0.9 | 0.02±0.01 | 45662 | 3555 | 10.8±0.8 |
| GL1713 | Boulder | 67 | 79.14 | -65.81 | 5.3 | 0.9998 | 40.261 | 0.214 | 10.5±0.7 | 0.02±0.01 | 35421 | 2451 | 8.3±0.6 |
| GL1714 | Boulder | 67 | 79.14 | -65.81 | 6.0 | 0.9999 | 40.088 | 0.214 | 11.3±1.4 | 0.02±0.01 | 38144 | 4846 | 9.0±1.1 |
| GL1715 | Pebbles | 58 | 79.03 | -67.80 | 0.7 | 0.9999 | 40.235 | 0.205 | 9.2±2.6 | 0.02±0.01 | 29660 | 8520 | 6.8±2.0 |
| GL1716 | Pebbles | 58 | 79.03 | -67.80 | 2.2 | 0.9999 | 40.196 | 0.212 | 20.9±5.6 | 0.02±0.01 | 69622 | 18782 | 16.2±4.4 |
| GL1717 | Boulder | 195 | 78.93 | -67.82 | 6.0 | 0.9981 | 40.114 | 0.443 | 9.8±0.5 | 0.02±0.01 | 68250 | 3283 | 14.0±0.7 |
| GL1718 | Boulder | 195 | 78.93 | -67.82 | 5.7 | 0.9981 | 40.012 | 0.210 | 64.8±10.5 | 0.02±0.01 | 215174 | 34752 | 44.4±7.3 |
| GL1719 | Boulder | 191 | 78.93 | -67.82 | 3.9 | 0.9997 | 40.031 | 0.207 | 19.7±0.8 | 0.02±0.01 | 64477 | 2743 | 13.0±0.5 |
| GL1720 | Boulder | 284 | 78.84 | -67.09 | 3.3 | 0.9993 | 40.159 | 0.215 | 23.3±0.9 | 0.02±0.01 | 78937 | 3097 | 14.3±0.6 |
| GL1721 | Boulder | 302 | 78.84 | -67.09 | 3.9 | 0.9983 | 40.038 | 0.208 | 28.9±0.7 | 0.2±0.05 | 94273 | 2377 | 17.0±0.4 |
| GL1722 | Boulder | 301 | 78.84 | -67.09 | 4.5 | 0.9983 | 40.016 | 0.205 | 30.3±0.7 | 0.2±0.05 | 99077 | 2296 | 17.9±0.4 |
| GL1723 | Boulder | 87 | 78.85 | -68.89 | 4.7 | 0.9989 | 40.039 | 0.203 | 12.8±0.5 | 0.2±0.05 | 40899 | 1501 | 9.4±0.3 |
| GL1724 | Boulder | 82 | 78.85 | -68.89 | 5.6 | 0.9992 | 40.024 | 0.206 | 12.3±0.4 | 0.2±0.05 | 39377 | 1370 | 9.1±0.3 |
| GL1725 | Boulder | 86 | 78.85 | -68.89 | 5.7 | 0.9991 | 40.006 | 0.203 | 53.8±0.9 | 0.2±0.05 | 172559 | 3088 | 40.2±0.7 |
| GL1732 | Boulder | 483 | 78.41 | -70.32 | 4.6 | 1 | 40.059 | 0.201 | 124.6±1.5 | 0.2±0.05 | 396868 | 5565 | 60.2±0.9 |
| GL1733 | Boulder | 483 | 78.41 | -70.32 | 5.6 | 1 | 39.648 | 0.201 | 168.7±1.9 | 0.2±0.05 | 542019 | 7114 | 83.4±1.1 |
| GL1734 | Boulder | 480 | 78.41 | -70.33 | 4.9 | 1 | 34.450 | 0.204 | 158.9±2.2 | 0.2±0.05 | 602773 | 9256 | 92.7±1.5 |
| GL1735 | Boulder | 482 | 78.41 | -70.33 | 5.3 | 1 | 40.007 | 0.208 | 147.0±1.5 | 0.2±0.05 | 484687 | 5982 | 74.3±0.9 |

[a] Carrier *Phe1602* (328.2±3.7 μg [9]Be/g) was used for preparation of sample GL1701-GL1703 and GL1706-GL1710. All other samples were prepared using carrier *Phe1603* (949.4±6.2 μg [9]Be/g).

[b] [10]Be ages were calculated using the online exposure age calculator formerly known as the CRONUS-Earth online exposure calculator v.3 (Balco et al., 2008), the Baffin Bay production rate (Young et al., 2013), and the St scaling scheme (Lal, 1991; Stone, 2000) under standard atmosphere. A rock density of 2.65 g cm[-3] was used and we assumed zero erosion. Samples were normalized to the Beryllium standard ICN-01-5-4, with a [10]Be/[9]Be value of 2.851 x 10[-12] (Nishiizumi et al., 2007) and blank corrected. [10]Be age uncertainties are reported as the 1σ internal uncertainty.



**Table 2:** Sample collection, [14]C isotopic information and resulting exposure ages for 4 boulders from Inglefield Land, north Greenland.

| Sample name | Elevation (m a.s.l.) | Latitude (°N) | Longitude (°W) | Shielding correction | Quartz (g) | $CO_2$ yield (µg) | $F^{14}C$ | $\delta^{13}C$ (‰) | $^{14}C/^{12}C_{total}$ ($10^{-14}$) | $^{14}C$ atoms blank corrected ($10^5$)[b] | $^{14}C$ ($10^5$ at g$^{-1}$) | $^{14}C$ exposure age (ka)[c] |
|---|---|---|---|---|---|---|---|---|---|---|---|---|
| GL1701 | 542 | 78.76 | 67.01 | 0.9999 | 3.9552 | 22.93 | 0.550 | -10.1 | 65.0±0.71 | 6.88±0.10 | 1.74±0.03 | 14.2±0.5 |
| GL1708 | 524 | 78.51 | 67.94 | 0.9998 | 3.9036 | 76.28 | 0.114 | -10.8 | 13.4±0.25 | 4.54±0.11 | 1.16±0.03 | 6.7±0.3 |
| GL1712 | 65 | 79.14 | 65.81 | 0.9998 | 3.9156 | 80.01 | 0.083 | -12.9 | 9.70±0.23 | 3.30±0.11 | 0.84±0.03 | 8.2±0.5 |
| GL1725 | 86 | 78.85 | 68.89 | 0.9991 | 3.8113 | 77.42 | 0.083 | -11.6 | 9.77±0.23 | 3.20±0.10 | 0.84±0.03 | 7.9±0.4 |

[a] Normalized to $\delta^{13}C$ of -25‰ VPDB and AD 1950

[b] All samples were blank corrected (0.589±0.052 $10^5$ [14]C atoms)

[c] [14]C ages were calculated using the online exposure age calculator formerly known as the CRONUS-Earth online exposure calculator v.3
(Balco et al., 2008), the west Greenland production rate (Young et al., 2014), and the Lm scaling scheme (Lal, 1991; Stone, 2000) under
standard atmosphere. A rock density of 2.65 g cm$^{-3}$ was used and we assumed zero erosion. [14]C age uncertainties are reported as the 1σ
analytical uncertainty.







**Table 3:** Sample collection information, radiocarbon ages and calibrated ages for marine molluscs collected at the margin of the Humboldt Glacier and wood fragments collected in front of the Hiawatha Glacier, north Greenland.

| Lab ID | Sample material | Latitude (ºN) | Longitude (ºW) | Elevation (m a.s.l.) | Age ($^{14}$C yr BP) | Age (95 % range) (cal. yr BP)[a] | Mean age (cal. yr BP±2σ)[a] |
|---|---|---|---|---|---|---|---|
| AAR-27511 | *Mya truncata* | | | | 2006±24 | 1321-1494 | 1400±44 |
| AAR-27512 | *Mya truncata* | | | | 1589±22 | 920-1050 | 983±35 |
| AAR-27513 | *Mya truncata* | | | | 3387±34 | 2937-3166 | 3052±60 |
| AAR-27514 | *Mya truncata* | | | | 2899±27 | 2345-2606 | 2466±70 |
| AAR-27515 | *Mya truncata* | | | | 3831±26 | 3501-3684 | 3593±44 |
| AAR-27516 | *Mya truncata* | | | | 1093±20 | 499-605 | 542±28 |
| AAR-27517 | *Mya truncata* | | | | 1415±23 | 736-891 | 812±41 |
| AAR-27518 | *Hiatella arctica* | 79.143 | 65.797 | 90 | 3413±25 | 2984-3181 | 3087±50 |
| AAR-27519 | *Hiatella arctica* | | | | 1988±28 | 1299-1474 | 1379±45 |
| AAR-27520 | *Hiatella arctica* | | | | 1195±23 | 553-662 | 613±30 |
| AAR-27521 | *Hiatella arctica* | | | | 2580±25 | 1979-2148 | 2066±45 |
| AAR-27522 | *Hiatella arctica* | | | | 1428±23 | 749-900 | 825±39 |
| AAR-27523 | *Hiatella arctica* | | | | 2318±33 | 1698-1884 | 1793±49 |
| AAR-27524 | *Hiatella arctica* | | | | 2248±22 | 1666-1865 | 1761±49 |
| AAR-27525 | *Astarte borealis* | | | | 1466±25 | 783-922 | 857±38 |
| 471815 | Wood | | | | 2260±30 | 2158-2346 | 2256±59 |
| 471816 | Wood | 78.830 | 67.133 | 193 | 1910±30 | 1741-1929 | 1854±37 |
| 471817 | Wood | | | | 2260±30 | 2158-2346 | 2256±59 |
| 471818 | Wood | | | | 5120±30 | 5751-5930 | 5846±59 |

[a] Radiocarbon ages were calibrated using OxCal v4.3 (Ramsey, 2009). The Marine13 calibration curve (Reimer et al., 2013) and a marine reservoir effect of 550 $^{14}$C yr (ΔR=150) (Mörner and Funder, 1990) were used for calibrating sample AAR-27511 to AAR-27525. For sample 675  471815-471818, the Intcall13 curve was used for calibration (Reimer et al., 2013).