# Peer review of "Glacial history of Inglefield Land, north Greenland from combined in-situ 10Be and 14C exposure dating"

_Climate of the Past, 2020_

## Short Comment (SC1) · 19 May 2020

I'm curious as to how the authors selected the modeling constraints in Figure 5. The authors recognize that sample GL1701, with an in situ 14C age of 14.2 ka, is likely influenced by inheritance and model an exposure-burial history that results in their measured 14C concentration. The authors state that they use a "known glacial history of the GrIS in north Greenland".

I think the younger constraint (6.7 ka) is from this manuscript, and this makes sense, but their model starts at 45 ka and it is unclear where that number comes from. The model also includes burial (no nuclide production) between 23 and 6.7 ka. The end

result is a modeled 14C concentration that is still below what they measured.

I think key here is that the 14C concentration at 23 ka (14C produced between 45 and 23 ka) is not quite at saturation, and this seems heavily dependent on starting the model at 45 ka. Why not start with a saturated concentration and then bury the sample between 23 and 6.7 ka? I guess using this simple exposure-burial scenario, this would mean starting your model at something closer to 50-53 ka. Seems reasonable and would end up with a 14C concentration closer to what you measured. If there is a solid reason for starting the model at 45 ka and not having the 14C concentration reach saturation prior to 23 ka, then how do you account for the remaining excess 14C?

---

## Author Comment (AC1) · 27 May 2020

The start of the model at 45 ka is based on the two studies cited in the text (Larsen et al., 2018; Søndergaard et al., 2019) from northeast and northwest Greenland. In these studies, radiocarbon ages of marine molluscs show that the GrIS in the study areas was behind its present day position during MIS3, starting as early as c. 42 cal. ka BP.

We have tried different model runs, including letting the initial concentration being at saturation (starting the model at 55-60 ka), but due to the short half-life of 14C, any concentration build-up prior to our constrained period only result in a very small increase in the final present-day concentration (c. 5%). It is therefore not exposure prior

to 45 ka, that is the main reason for the remaining "inheritance".

We are therefore unable to explain the excess 14C in our sample from the known glacial history, which we also state in the paper:

"However, as we only have one data point and the simulation is incapable of fully reaching the measured concentration we cannot make any firm conclusions on the timing of prior exposure of the sample and the implications for the ice sheet history."

―――――――――――――――――――――

---

## Referee Comment (RC1) · Yarrow Axford (Referee) · 29 Jun 2020

This paper presents new 10Be and in situ 14C constraints on the timing of early to middle Holocene deglaciation of Inglefield Land in northwest Greenland, and 14C ages from reworked organic materials that record a period in the middle to late Holocene when the Greenland Ice Sheet in the study area was smaller than present. The combination of 10Be and in situ 14C reveals extensive nuclide inheritance in the region, indicating past cold-based ice cover / mimimally erosive ice, especially on highlands. The paper thoughtfully integrates all data types to reconstruct the ice sheet margin history in an area where more data are badly needed. And I appreciate the review of

prior work to piece together a broader picture of regional glacial history. Altogether, the paper documents some regional coherence as well as complex spatial variations in the timing of glacier margin changes, and those patterns should ultimately – in concert with future work to flesh out the paleoclimate and/or glacial history in even greater detail – help our community understand key aspects of ice sheet dynamics. I enthusiastically recommend this study for publication after minor revisions. It takes a thorough, multi-method approach to fleshing out the glacial history of a poorly known sector of the Greenland Ice Sheet.

My most significant suggestion is to better describe the morphostratigraphic contexts of the 14C-dated organic materials, and in the case of the wood fragments the rationale for inferring that they derive from inboard of the modern ice sheet margin. I think the link between minimum ice sheet extent in the middle Holocene and the inferred driver of ocean climate (vs. atmospheric), could also be further considered and further justified.

Detailed comments:

Methods and Tables 1/2: What were the lithologies of the boulders sampled for 10Be? Were they consistent with the local bedrock (or likely far-traveled from inland under the ice sheet)? (And in Results, any pattern of different lithologies among the oldest vs youngest 10Be ages, ie degree of inheritance?) Or is everything uniformly granitoid/gneiss with local vs exotic provenance impossible to pin down?

Please describe further the morphostratigraphic contexts of the dated wood fragments. Were they exposed on the surface of the meltwater plain, coming out in meltwater right at the ice front, or found buried in an outcrop of river deposits? Any evidence for the species of the "wood"? It would be useful to include any information that rules out or argues against these materials having been exhumed by water or wind from a nearby soil (instead of excavated by ice inboard of the present-day ice sheet margin, as is inferred). This possibility should be discussed in the Results and/or Discussion as well.

There is a brief description of context for the 14C-dated molluscs (on and within di-amicts), but it appears in Results and I suggest putting this fundamental sampling information in Methods.

Line 273: "the ice margin reached its present-day extent at Delta Sø c. 10.1 ka" The age of 10.1 ka is actually the basal age from Wax Lips Lake, which is indeed the best constraint on when ice in that region reached its modern extent because WLL is situated only ∼2 km from the modern ice margin (McFarlin et al 2018 PNAS, discussed in Axford et al. 2019). Suggest changing "Delta So" in this sentence to "Wax Lips Lake" and citing McFarlin (and add WLL to Fig 8a if needed).

Line 284: "Farther north in the Thule area and around Qaanaaq, mosses from a local ice cap and subfossil plants from the GrIS show a smaller ice extent before c. 3.3 cal. ka BP (Farnsworth et al., 2018; Axford et al., 2019. . ." Just a note that Axford et al. also find the North Ice Cap was smaller than present for most of the Holocene, as reflected in your Fig 8c, and that seems to contrast with the wording here.

Line 299: I think it is debatable whether the early Holocene peak warmth in NW Green-land was "earlier than in the rest of Greenland." What is the evidence for later onset of warmth everywhere else? There is some evidence for early warmth in the east, includ-ing from Renland ice cap (which unlike most of the central Greenland ice core records and I think the very nice Buizert work, is elevation-corrected). Suggest just removing this statement that generalizes across all of Greenland, and keeping your discussion focused on the evidence for timing of warmth in the Nares Strait region vs a bit further south in NW Greenland, as you already mostly do. Also, given the dearth of diverse ev-idence for atmospheric temperatures themselves in the Nares Strait region, it would be interesting to see a more fleshed-out discussion of the possible climate interpretations of the ice sheet history. Is it possible that the ice margin history is somehow compatible with early Holocene peak temperatures (more sensitive to ocean temperatures, longer lag in ice sheet equilibrium, more sensitive to precip??), or does the ice margin history truly preclude that?

Figure 9: I don't think I've seen ice margin histories summarizes in quite this way visually before, and I really like it! Useful way to represent the data across a range of studies.

General point on the Discussion: One major conclusion of the cited Reusche study nearby is that the ice margin responded to cold events ~9.3 or 8.2 ka, interrupting rapid retreat in the early Holocene. That should probably be acknowledged and discussed at least briefly. Do the new data generated in the current study add to or modify that picture?

Discussion, ~line 310 etc: While invoking ocean temperatures to drive mid-Holocene minimum ice extent, it is also worth noting that many paleotemperature proxies from Greenland and Agassiz indicate that air temperatures were elevated above those of the late Holocene and even the 20th Century well into the middle Holocene. Could the minimum ice extent in the mid Holocene alternately represent a lagged equilibrium with warmer-than-20th C temperatures?

---

## Referee Comment (RC2) · Anonymous Referee #2 · 7 Jul 2020

The paper by Søndergaard et al. provides new CRN exposure ages (10Be, 14C) from erratic boulders and pebbles and radiocarbon ages from wood fragments to constrain the glacial history in Inglefield Land in northern Greenland. Based on their data, the authors conclude that that the glacial history in Inglefield Land commenced around 8.5 ka along the western margin and around 7.9 ka in the central part and reached is present position in the central part of the Inglefield by 6.9 ka. An overview of the Northern-Northwestern Greenland Ice Sheet history was also summarized as part of the work, including potential climate forcings.

Overall, this is a very nice study and will make a nice addition to the literature. Like the

[Figure]

two other reviewers, I have similar questions regarding Figure 5 and how it was constructed and some more detailed questions about the some of the data (e.g. lithologies of boulders). Rather than repeat their questions, some of which I also had, I have provided my figure related questions and a few other comments that I hope the authors will address prior to publication. Otherwise this is a very nice, succinct paper that I was very pleased to read and really liked. Great work folks!

Figure 3: It would be useful if the authors included the uncertainties on the figure. Perhaps just including the average 10Be and 14C uncertainty in the legend would suffice. This is important to readers who may not encounter these types of data often need some baseline to who precise the measurements can be. Authors choice on this one since I'm only suggesting it.

Figure 4: I'm not sure I find this figure particularly useful. Does it provide anything more that the table doesn't already provide the reader?

Figure 5: I have the same sentiments as Nicolas on this, so will let you address his comment.

Figure 7b: How does this work compare with the raised beach records from Bennike, 2002 or the modeling work from Lecavalier et al. 2017? The schematic in part b of this figure is interesting and makes me wonder how it might compare to those relative sea level curves and ice margin reconstructions. It might be worth mentioning something in this regard within the text.

Figure 8b: I like these figures that the authors provide. However, it isn't clear to me how they derive some of these numbers. For instance, in Washington Land to the north of their site the authors provide an outer coastal retreat around 9.0 ka and present day ice margin around 8.6 ka. Based on my read of the Ceperley et al. 2020 paper, it seems like the ice margin was at Crozier Island at 8.5 ka and within the interior around 7.6 ka based on taking the youngest 10Be ages. These ages are consistent with what is being found in Inglefield Land and would indicate to me that over this entire

[Figure]

area in the Northwest that the glaciers were largely acting in unison with no significant leads/lags. Perhaps the authors have recalculated these ages which is the reason for the discrepancy but regardless this should be addressed and explained assuming this is the case.

Lines 209-210: I'm not sure how you get inheritance for 14C in this region but I agree with the authors that this age seems unrealistic. Based on Figure 5, the authors have suggested that during MIS 3 this location was ice free. This is a really interesting hypothesis and I think the authors should explain how this might be possible (e.g. climatically, glaciologically) given most people typically don't think of MIS 3 as that much different than the LGM, yet the authors Figure 5 would make MIS 3 seems similar to the present day. More should be said here since this hypothesis has some implications for what the climate might be like in the past and the authors could weigh in on it.

—————————————————

---

## Author Response (AR2)

**Editor Decision: Publish subject to technical corrections** (22 Sep 2020) by Irina Rogozhina
Comments to the Author:
Dear authors,

With this email I would like to confirm that your revised manuscript is nearly ready for publication. I feel that your discussion of the large-scale versus regional drivers of the Holocene changes in NW and N Greenland could benefit from a more detailed overview of the modeling studies and ocean proxy data. However, I also feel that it might trigger a shift in the study's focus from the empirical evidence to the general circulation patterns.

Before I recommend your article for publication, I would like your text to undergo a careful proofread. I have spotted a newly introduced sentence that is too long and difficult to follow (lines 318 - 321) and a few mistakes/misprints (e.g., lines 115, 244, 375).
We have re-written the long sentence and divided it into smaller sentences, for clarity.

We have further done a careful proofread and made corrections for misprints/mistakes where necessary.

All corrections are highlighted in the track-n-trace document.

I look forward to receiving the final version of the manuscript

Kind regards,

Irina

**Editor Decision: Publish subject to minor revisions (review by editor)** (23 Aug 2020) by Irina Rogozhina
Comments to the Author:
Dear authors,

I have now gone through your responses to the reviews and the short comment and in most cases I agree with your strategies for addressing the concerns raised by the reviewers. I therefore encourage you to address these comments as fully as possible following and expanding upon your proposed strategies.
Thank you. We have elaborated more on all comments and made changes accordingly in the manuscript.

In addition, I would like you to include the ensemble of model experiments (and their setup) that you are describing in your reply to the short comment (by Nicolas Young) in the form of supplementary materials.
We have decided to change figure 5, so it now includes the original figure (a) as well as a model run (b) where we start the model at 60 ka, as describe in our response to Nicolas Young. After your comment, we thought it would be good for the reader to have both scenarios in the main text, as it is an important point in the conclusion as to why the sample is affected by inheritance. We have changed the caption for figure 5 accordingly and also added a modified version of our response to Nicolas Young in the main text in section 5.1, where we describe why we have chosen the age constraints for our model the way we have.

I agree with the authors that Figure 4 is a useful visualization of the table contents and should therefore be retained.

The figure remains in the manuscript as it was.

The common line in the reviewers' comments is the need for a better integration between the results of this study and the interpretation of paleoclimate evidence related to the major drivers of the reconstructed ice sheet behavior. Hence, I would like to see a more detailed analysis and discussion of this in the revised manuscript.

We have added to the discussion in various sections addressing the climate-ice interactions that were questioned by the reviewers. Especially, section 5.4 has been expanded specifically in accordance to the reviewer comments. Specifically, we comment on the effect from the retreating Inuitian Ice Sheet on ice retreat in northern Greenland, which might have delayed this and further we have also added information on the effect from rising SSTs on the ice sheet that we see in northwest Greenland.

Good luck with the implementation of the reviews. I look forward to seeing the new version of the manuscript.

Kind regards,

Irina

Interactive comment on "Glacial history of
Inglefield Land, north Greenland from combined
in-situ 10Be and 14C exposure dating" by Anne
Sofie Søndergaard et al.

Yarrow Axford (Referee)
axford@northwestern.edu

This paper presents new 10Be and in situ 14C constraints on the timing of early to middle
Holocene deglaciation of Inglefield Land in northwest Greenland, and 14C ages
from reworked organic materials that record a period in the middle to late Holocene
when the Greenland Ice Sheet in the study area was smaller than present. The combination
of 10Be and in situ 14C reveals extensive nuclide inheritance in the region,
indicating past cold-based ice cover / mimimally erosive ice, especially on highlands.
The paper thoughtfully integrates all data types to reconstruct the ice sheet margin
history in an area where more data are badly needed. And I appreciate the review of prior work
to piece together a broader picture of regional glacial history. Altogether, the
paper documents some regional coherence as well as complex spatial variations in the
timing of glacier margin changes, and those patterns should ultimately – in concert with
future work to flesh out the paleoclimate and/or glacial history in even greater detail –
help our community understand key aspects of ice sheet dynamics. I enthusiastically
recommend this study for publication after minor revisions. It takes a thorough, multimethod
approach to fleshing out the glacial history of a poorly known sector of the
Greenland Ice Sheet.

My most significant suggestion is to better describe the morphostratigraphic contexts
of the 14C-dated organic materials, and in the case of the wood fragments the rationale
for inferring that they derive from inboard of the modern ice sheet margin. I think the
link between minimum ice sheet extent in the middle Holocene and the inferred driver of
ocean climate (vs. atmospheric), could also be further considered and further justified.
The above has been addressed in the following individual comments and changes have been
incorporated into the manuscript.

Detailed comments:

Methods and Tables 1/2: What were the lithologies of the boulders sampled for 10Be?
Were they consistent with the local bedrock (or likely far-traveled from inland under
the ice sheet)?

Answer: We have added a couple of lines in the Methods section about the lithology and our
interpretation of transport distance.

"The lithology of boulders sampled were all granite or gneiss, matching the lithology in and
around Inglefield Land, but due to the placement on bedrock and moraines we assume the
boulders to be transported some distance before deposition."

 (And in Results, any pattern of different lithologies among the oldest
vs youngest 10Be ages, ie degree of inheritance?) Or is everything uniformly granitoid/
gneiss with local vs exotic provenance impossible to pin down?

Answer: Everything is uniformly granitoid/gneiss. We have looked into possible patterns between the lithology and inheritance, especially Feldspar (Al) content in the samples, but there is no clear pattern between ages and lithology. We have added a comment in the results section (4.1) about this.

Please describe further the morphostratigraphic contexts of the dated wood fragments. Were they exposed on the surface of the meltwater plain, coming out in meltwater right at the ice front, or found buried in an outcrop of river deposits?

Answer: The wood fragments were collected together with samples presented in recent studies concerning the Hiawatha Impact crater (Kjær et al., 2018; Garde et al., 2020). We have added more information on the sample site in the methods section 3.2.

Any evidence for the species of the "wood"?

Answer: It has not been possible to identify the dated wood found in front of the Hiawatha Glacier.

It would be useful to include any information that rules out or argues against these materials having been exhumed by water or wind from a nearby soil (instead of excavated by ice inboard of the present-day ice sheet margin, as is inferred). This possibility should be discussed in the Results and/or Discussion as well.

Answer: We have commented on this in the results section 4.2, where we argue for our interpretation of the wood fragments having originated from underneath the glacier.

"As the wood fragments were retrieved right in front of the Hiawatha Glacier, the wood fragments are more likely to have originated from underneath the glacier than being transported from a nearby soil to the sample site by wind or water. We therefore believe these ages to constrain a time when the glacier was smaller than its present-day extent."

There is a brief description of context for the 14C-dated molluscs (on and within diamicts), but it appears in Results and I suggest putting this fundamental sampling information in Methods.

Answer: This information has been moved so it now appears for the first time in the methods section.

Line 273: "the ice margin reached its present-day extent at Delta Sø c. 10.1 ka" The age of 10.1 ka is actually the basal age from Wax Lips Lake, which is indeed the best constraint on when ice in that region reached its modern extent because WLL is situated only _2 km from the modern ice margin (McFarlin et al 2018 PNAS, discussed in Axford et al. 2019). Suggest changing "Delta So" in this sentence to "Wax Lips Lake" and citing McFarlin (and add WLL to Fig 8a if needed).

Answer: The sentence and reference have been changed as suggested and the location of Wax Lips Lake has been added to figure 8a.

Line 284: "Farther north in the Thule area and around Qaanaaq, mosses from a local

ice cap and subfossil plants from the GrIS show a smaller ice extent before c. 3.3 cal. ka BP (Farnsworth et al., 2018; Axford et al., 2019: : :" Just a note that Axford et al. also find the North Ice Cap was smaller than present for most of the Holocene, as reflected in your Fig 8c, and that seems to contrast with the wording here.

Answer: The 3.3 cal. ka BP is linked to the study around Qaanaaq - we have re-phrased the sentence for clarity so it now reads: Farther north, mosses from a local ice cap and subfossil plants from the GrIS show a smaller than present-day ice extent before c. 3.3 cal. ka BP around Qaanaaq and throughout most of the Holocene until c. 1850 AD in the Thule area (Farnsworth et al., 2018; Axford et al., 2019; Søndergaard et al., 2019).

Line 299: I think it is debatable whether the early Holocene peak warmth in NW Greenland was "earlier than in the rest of Greenland." What is the evidence for later onset of warmth everywhere else? There is some evidence for early warmth in the east, including from Renland ice cap (which unlike most of the central Greenland ice core records and I think the very nice Buizert work, is elevation-corrected). Suggest just removing this statement that generalizes across all of Greenland, and keeping your discussion focused on the evidence for timing of warmth in the Nares Strait region vs a bit further south in NW Greenland, as you already mostly do.
Answer: We have deleted the statement as suggested.

Also, given the dearth of diverse evidence for atmospheric temperatures themselves in the Nares Strait region, it would be interesting to see a more fleshed-out discussion of the possible climate interpretations of the ice sheet history. Is it possible that the ice margin history is somehow compatible with early Holocene peak temperatures (more sensitive to ocean temperatures, longer lag in ice sheet equilibrium, more sensitive to precip??), or does the ice margin history truly preclude that?

Answer: We have added a part in section 5.4 about the possible effects from the Innuitian Ice Sheet on the western north GrIS during early Holocene. Following these lines, we already commented on the effects of rising temperatures and the opening of Nares Strait on the north GrIS margin and its retreat.

Figure 9: I don't think I've seen ice margin histories summarizes in quite this way visually before, and I really like it! Useful way to represent the data across a range of studies.

Answer: Thank you!

General point on the Discussion: One major conclusion of the cited Reusche study nearby is that the ice margin responded to cold events _9.3 or 8.2 ka, interrupting rapid retreat in the early Holocene. That should probably be acknowledged and discussed at least briefly. Do the new data generated in the current study add to or modify that picture?

Answer: In section 2, Study site and previous work, we briefly mention the study of Reusche et al and the possible stillstand of the ice sheet at 8.2 ka. Further, in section 5.2, we discuss the exposure ages from Humboldt Glacier in connection to the study from Reusche et al: Farthest north in Inglefield Land, at the southern flank of the Humboldt Glacier the ice margin reached its present-day extent already by c. 8.2 ka (Fig. 7a). This age is consistent

with the [10]Be chronology from the northern flank of Humboldt Glacier where a moraine a few hundred meters outside the LIA moraine was abandoned c. 8.3 ka (Reusche et al., 2018). We have therefore not added anything further to the text about this.

Discussion, _line 310 etc: While invoking ocean temperatures to drive mid-Holocene minimum ice extent, it is also worth noting that many paleotemperature proxies from Greenland and Agassiz indicate that air temperatures were elevated above those of the late Holocene and even the 20th Century well into the middle Holocene. Could the minimum ice extent in the mid Holocene alternately represent a lagged equilibrium with warmer-than-20th C temperatures?

Answer: We believe that the rising sea surface temperatures in middle Holocene were the main driver of the smaller than present-day extent of the ice sheet in northwest Greenland. But it is likely that warming sea surface temperatures boosted a trend already happening from high atmospheric temperatures. We have added a few lines to section 5.4.

Interactive comment on "Glacial history of
Inglefield Land, north Greenland from combined
in-situ 10Be and 14C exposure dating" by Anne
Sofie Søndergaard et al.

Anonymous Referee #2

The paper by Søndergaard et al. provides new CRN exposure ages (10Be, 14C) from
erratic boulders and pebbles and radiocarbon ages from wood fragments to constrain
the glacial history in Inglefield Land in northern Greenland. Based on their data, the
authors conclude that that the glacial history in Inglefield Land commenced around
8.5 ka along the western margin and around 7.9 ka in the central part and reached
is present position in the central part of the Inglefield by 6.9 ka. An overview of the
Northern-Northwestern Greenland Ice Sheet history was also summarized as part of
the work, including potential climate forcings.

Overall, this is a very nice study and will make a nice addition to the literature. Like thetwo
other reviewers, I have similar questions regarding Figure 5 and how it was constructed
and some more detailed questions about the some of the data (e.g. lithologies
of boulders). Rather than repeat their questions, some of which I also had, I have provided
my figure related questions and a few other comments that I hope the authors
will address prior to publication. Otherwise this is a very nice, succinct paper that I was
very pleased to read and really liked. Great work folks!

Answer: Thank you!

Figure 3: It would be useful if the authors included the uncertainties on the figure.
Perhaps just including the average 10Be and 14C uncertainty in the legend would
suffice. This is important to readers who may not encounter these types of data often
need some baseline to who precise the measurements can be. Authors choice on this
one since I'm only suggesting it.

Answer: We have added range and average of age uncertainties in the figure caption and refer
readers to Table 1 if they want more specific information on individual samples.

Figure 4: I'm not sure I find this figure particularly useful. Does it provide anything more
that the table doesn't already provide the reader?

Answer: The figure does not provide more information than the table does, but we find it
useful to represent data in a figure for better overview and clarity of the age distribution
between the two glaciers and two sample materials.

Figure 5: I have the same sentiments as Nicolas on this, so will let you address his
comment.

Answer: We have decided to change figure 5, so it now includes the original figure (a) as
well as a model run (b) where we start the model at 60 ka, as describe in our response to
Nicolas Young. We thought it would be good for the reader to have both scenarios in the
main text, as it is an important point in the conclusion as to why the sample is affected by

inheritance. We have changed the caption for figure 5 accordingly and also added a modified version of our response to Nicolas Young in the main text in section 5.1, where we describe why we have chosen the age constraints for our model the way we have.

Figure 7b: How does this work compare with the raised beach records from Bennike, 2002 or the modeling work from Lecavalier et al. 2017? The schematic in part b of this figure is interesting and makes me wonder how it might compare to those relative sea level curves and ice margin reconstructions. It might be worth mentioning something in this regard within the text.

Answer: The schematic part in Figure 7b is partly based on our results and partly on previous work (some of it used in Bennike (2002), which we mention in the discussions section 5.2. In this section we specifically discuss our result in relation to previous studies and how to find the best fit from both regarding the ice sheet history.

Figure 8b: I like these figures that the authors provide. However, it isn't clear to me how they derive some of these numbers. For instance, in Washington Land to the north of their site the authors provide an outer coastal retreat around 9.0 ka and present day ice margin around 8.6 ka. Based on my read of the Ceperley et al. 2020 paper, it seems like the ice margin was at Crozier Island at 8.5 ka and within the interior around 7.6 ka based on taking the youngest 10Be ages. These ages are consistent with what is being found in Inglefield Land and would indicate to me that over this entire area in the Northwest that the glaciers were largely acting in unison with no significant leads/lags. Perhaps the authors have recalculated these ages which is the reason for the discrepancy but regardless this should be addressed and explained assuming this is the case.

Answer: For the outer coast estimate at 9.0 ka, we have taken the mean which Ceperly et al. 2020 provide from Cozier Island and Joe Island. We find this mean representative for an outer coast deglaciation age of Washington Land: "The Holocene exposure ages from Crozier Island and Joe Island within Nares Strait have a mean of $9.0 \pm 1.1$ ka (n . 7; 1-s)."
For the inner coast estimate at 8.6 ka, we chose the average for "widespread ice sheet retreat" in Washington Land as stated by the authors. We acknowledge that a better estimate for when the ice what at its present-day extent might be the estimate from the authors at 6.9 ka for when "widespread glacial ice was absent". We have therefore changed the estimate for the inner coast deglaciation accordingly in the text (section 5.3) and Figure 8b.

Lines 209-210: I'm not sure how you get inheritance for 14C in this region but I agree with the authors that this age seems unrealistic. Based on Figure 5, the authors have suggested that during MIS 3 this location was ice free. This is a really interesting hypothesis and I think the authors should explain how this might be possible (e.g. climatically,glaciologically) given most people typically don't think of MIS 3 as that much different than the LGM, yet the authors Figure 5 would make MIS 3 seems similar to the present day. More should be said here since this hypothesis has some implications for what the climate might be like in the past and the authors could weigh in on it.

Answer: Following our conclusion of the $^{14}$C age being affected by inheritance we comment on the possibility of a smaller than present day extent of the GrIS during MIS3, which fits with other studies from northern Greenland, which have concluded the same: "This scenario

is to some degree consistent with other studies in northern Greenland that suggest a restricted GrIS during MIS 3 (Larsen et al., 2018; Søndergaard et al., 2019) and a late coalescence of the GrIS and Inuitian Ice Sheet around 22 cal. ka BP (England, 1999)". In the following lines we state that we can't make any firm conclusion due to the lack of data, which is why we have not elaborated more on the specific climatic conditions which would cause a MIS3 comparable to our present-day situation.

[revised manuscript text omitted]